# Instrumenting *Polyodon spathula* (Paddlefish) Rostra in Flowing Water with Strain Gages and Accelerometers

**DOI:** 10.3390/bios10040037

**Published:** 2020-04-11

**Authors:** Clayton R. Thurmer, Reena R. Patel, Guilermo A. Riveros, Quincy G. Alexander, Jason D. Ray, Anton Netchaev, Richard D. Brown, Emily G. Leathers, Jordan D. Klein, Jan Jeffrey Hoover

**Affiliations:** 1US Army Engineer Research & Development Center, Information Technology Lab, 3909 Halls Ferry Road, Vicksburg, MS 39180, USA; Reena.R.Patel@erdc.dren.mil (R.R.P.); Guillermo.A.Riveros@erdc.dren.mil (G.A.R.); Quincy.G.Alexander@erdc.dren.mil (Q.G.A.); Jason.D.Ray@erdc.dren.mil (J.D.R.); Anton.Netchaev@erdc.dren.mil (A.N.); Richard.D.Brown@erdc.dren.mil (R.D.B.); Emily.G.Leathers@erdc.dren.mil (E.G.L.); Jordan.D.Klein@erdc.dren.mil (J.D.K.); 2US Army Engineer Research & Development Center, Environmental Lab, 3909 Halls Ferry Road, Vicksburg, MS 39180, USA; Jan.J.Hoover@usace.army.mil

**Keywords:** strain gage, accelerometer, instrumentation, ARM M0, Low-SWaP, printed circuit board (PCB), *Polyodon spathula*, bio-inspired materials, bio-inspired structures, in vivo

## Abstract

The prominent rostrum of the North American Paddlefish, supported by a lattice-like endoskeleton, is highly durable, making it an important candidate for bio-inspiration studies. Energy dissipation and load-bearing capacity of the structure from extreme physical force has been demonstrated superior to that of man-made systems, but response to continuous hydraulic forces is unknown and requires special instrumentation for in vivo testing on a live fish. A single supply strain gage amplifier circuit has been combined with a digital three-axis accelerometer, implemented in a printed circuit board (PCB), and integrated with the commercial-off-the-shelf Adafruit Feather M0 datalogger with a microSD card. The device is battery powered and enclosed in silicon before attachment around the rostrum with a silicon strap "watch band." As proof-of-concept, we tested the instrumentation on an amputated Paddlefish rostrum in a water-filled swim tunnel and successfully obtained interpretable data. Results indicate that this design could work on live swimming fish in future in vivo experiments.

## 1. Introduction

An elongated beak-like snout, or rostrum, is prominent among several families of fishes including sawfish (Pristidae), swordfish (Xiphiidae), and marlins (Istiophoridae), but it is especially pronounced in the North American Paddlefish, *Polyodon spathula* (Polyodontidae) [1]. The Paddlefish, a “living fossil” with closely-related ancestors alive in the Cretaceous Era, has a rostrum unlike any other fish, supported by a lattice-like skeleton of cartilage made up of star-shaped elements called stellate “bones” [2]. The rostrum is unusually large and variable in shape [3,4]. It has multiple vital functions, as a sensory antennae detecting vibratory and electrical signals [5,6,7] and as a locomotor structure providing lift and enhanceed swim speed [8,9].

Paddlefish can live for 30–60 years [10] and migrate extraordinary distances (up to 2000 km) through structurally and hydraulically complex environments [11,12]. As a result, their rostra experience decades of mechanical and hydraulic abrasion and sometimes trauma. However, the rostrum is highly durable and physical damage is uncommon. Based on hundreds of field surveys throughout their range of the Mississippi and Missouri River tributaries and Mobile Bay drainage basin, most populations of Paddlefish show no damaged or missing rostra, a low percentage (<10%) of damaged rostra, or minor, non-catastrophic damages to the rostra [13,14]. This is remarkable given the high speed waters through which Paddlefish regularly swim and the highly dynamic means by which they swim, with frequent dives and surfacings, often at sufficient speed to breach the surface of the water [14]. The durability of the Paddlefish rostrum, given the challenges of its physical environment, therefore, makes it an important model for bio-inspired materials and structures.

Most studies of the Paddlefish rostrum have been conducted by applying physical forces at specific points on the rostrum or to parts of the rostrum and then observing structural responses empirically or computationally. Such an approach is compatible with tests of other lattice-like biological structures such as turtle shells [15] and can be highly informative. Allison et al. [16] demonstrated variation in elasticity of individual bones depending on their spatial location within the rostrum; Deang et al. [17] quantified stress and strain of the mid-line cartilage and showed that it was not spatially dependent. Modelling efforts include a series of preliminary numerical experiments carried out by Riveros et al. [18] that revealed that the rostrum has superior energy dissipation and load bearing capacity as compared to man-made engineered systems. In other previous works, Patel et al. [19] studied stress flow patterns on the rosta of Paddlefish using a finite element and flow network approach to create a flow network graph. This approach was validated in their work [19,20,21] using classical approaches. Recently, Riveros et al. [22] developed finite-element models (FEM) to evaluate the mechanical behavior of the rostrum.

Responses to hydrodynamic forces are unstudied, but those data would provide important information on resiliency and stability of the rostrum (or similar bio-inspired structures) in moving water. Paddlefish swim at high speeds for extensive distances [8,11,14]. Allen and Riveros [9] demonstrated that lift generated by the rostrum was comparable to or exceeded that for man-made hydrofoils. Patel and Riveros [23] conducted fluid–structure interaction analysis on the rostrum. Knowing how the rostrum responds to orientation in the water column and variation in water velocity is crucial for the development of bio-inspired structures that must be positioned or operated underwater (e.g., pilings and submersibles). Estimates of strain (from gages) and vibration (from accelerometers) are needed, ideally from a captive, freely-swimming Paddlefish monitored in a swim tunnel [24], but first instrumentation must be designed or adapted that can be attached to the rostrum, operate while submerged and moving through water at various speeds, and provide interpretable data. In this paper, we present specifications for such instrumentation, a protocol for implementation, and a proof-of-concept experiment demonstrating efficacy. These are critical first-steps for eventual in vivo studies and are designed according to specific goals of in vivo studies.

A review of in vivo strain measurements reveals the widespread use of bone implanted resistance strain gages as early as 1985 [25]. Wire or foil resistance strain gages were implanted directly onto the bone surfaces permitting precise determination of in vivo bone deformations during functional activities [25]. In a more recent example from 2005, Milgrom et al. [26] implanted resistance strain gages using strain-gaged staple (SGS) in the human tibia with data acquisition components for four channels fitting into a military spec backpack. This experiment used strain-gaged staples (SGS) made from bone staples 16 × 15 mm in size with 350 Ω strain gage bonded to the undersurface [26]. Each SGS was wired as a quarter Wheatstone bridge connected to an unidentified portable four-channel amplifier [26]. The output signals of this conditioning amplifier were recorded on an FM analogue cassette recorder with playback by a separate unit connected to a PC and digitized at 400 Hz [26].

Importantly, precedent for comparison of in vivo strain measurements for validation of numerical models is discussed in the case of the alligator (*Alligator mississippiensis*) cranium during biting. Metzger et al. [27] have compared Beam Theory and Finite-Element Analysis with in vivo bone strain data. Clear differences were observed between the data extrapolated from the beam model, extracted from the FE model, and the recorded in vivo strains [27]. This was notable with regard to both the absolute strain magnitudes and the patterns of strain gradients across the skull [27]. The data signals were conditioned and amplified on a Vishay 2100 bridge amplifier (Malvern, PA, USA) and acquired at 1 KHz through a National Instruments data acquisition (DAQ) board (Austin, TX, USA) run by a MiDAS data acquisition software package [27]. A preliminary strain gage data logging design for the paddlefish rostrum was presented and validated against a calibrated system in a prior study [28]; however, this manuscript expands on this earlier work to address integration with an accelerometer, waterproofing, attachment to the rostrum, and additional measurement validation procedures. In addition, preliminary results of recording strain and acceleration from a rostrum are included under simulated swimming conditions.

## 2. Approach and Design

The literature confirms the approach to use a resistive strain gage in vivo [25,26,27] for validation of numerical results [27] but does not offer an instrumentation solution [26,27] portable to the distinct aquatic use-case for Paddlefish rostra in motion. Therefore, the instrumentation challenge is framed as a low- size, weight, and power (low-SWaP) system design and integration. The approach is to develop a small measurement and data logging solution which can be physically attached to the fish so that the swimming motion is not encumbered by wires. Figure 1 shows a conceptual placement of instruments on the rostrum of a paddlefish which was first discussed in a previous work [28]. The paddlefish head is shown for reference with the device near the area between the eyes. Tests may be performed on an amputated rostrum or on the rostrum of a live and swimming fish. The following subsections address the implementation details of the measurement device, the waterproof enclosure and attachment mechanism, and strain gage attachment to the rostrum.

### 2.1. Measurement

The notional instrumentation design concept representing the integrated system is shown in Figure 2. The Adafruit Feather M0 Adalogger (Adafruit, New York, NY, USA) was chosen as a microcontroller in this design with the integration of a measurement peripheral board first published in [28]. The microcontroller uses a microSD card for storing the data collected from the analog to digital converter (ADC) to be collected post-experiment. The peripheral board conditions and amplifies the strain gage signal [28]. Additional components have now been included on this board for three-axis digital acceleration ±16 g measurement using Serial Peripheral Interface (SPI) connections to the microcontroller.

Towards the low-SWaP design goal, the form factor of 51 mm × 23 mm × 19 mm is achieved with two strain measurements. The board layout design was a significant challenge of the project. Figure 3 shows the resulting device design with a scale reference included. The side view of Figure 3b shows that the header pins are soldered directly between the boards to save space. This is opposed to a conventional approach using female headers which would result in a thicker device. The battery is sandwiched between the boards and only the wire to the JST jack is visible. Importantly, the sensor outputs have been validated and calibrated within the integrated system. The successful calibration of the strain gages and accelerometer follows in the respective subsections. Finally, the power consumption is discussed.

#### 2.1.1. Strain

The output of strain gages is a change in resistance that is representative of a change in strain. A strain gage is typically measured as a Wheatstone bridge resistor divider and must be amplified such that the ADC resolution can detect the small changes in resistance for microstrain (μϵ). This study uses two types of gages. The model 1-LY41-3/1000 from Micro-Measurements (Raleigh, NC, USA) with k-alloy foil in combination with a tough flexible polymide backing. In addition, the model EK-06-060CD-10C/P 1-LY41-3/1000 from Hottinger Baldwin Messtechnik GmbH (HBM, Darmstadt, Germany) is made of constant measuring grid material with polyimide carrier foil. Both are chosen for 1000 Ω nominal resistance.

The Single Supply Strain gage Amplifier Circuit from Analog Devices reference design [29] has been implemented in PCB with form factor integration to the Adafruit Feather M0 datalogger [30] towards the measurement goal which is described in detail in [28]. Similarly, the algorithm to convert the measured voltage to strain and the respective software considerations of the strain measurement is discussed in [28]. A testing environment was established to validate the device by attaching two strain gages adjacent each other on a cantilevered steel plate. The steel plate was affixed to a workbench with clamps. A Reference Calibration technique was used to validate the measured strain output of the device to a known and calibrated system by comparing adjacent gages under arbitrary manipulation of the plate as published in [28].

A Load Calibration technique based on Liimatta [31] was subsequently used to show linearity of both device channels using the adjacent strain gages with respect to known loads. Shunt Calibration is discussed in the literature [32,33,34] but was not used due to lack of available shunt resistors with the necessary values. The two adjacent gages of the test plate are plugged into the two channels of the device and load calibrated by successively adding weights, as shown in Figure 4. Figure 5 shows the recorded values converted to strain versus time. The respective offsets are accounted for using the initial conditions as the zero reference. The signal noise is attributed to mechanical oscillations. In order to reduce the impact of these oscillations on the analysis, the strain is averaged across the approximate duration of each step. Figure 6 shows the two channels of the device averaged adjacent strain versus load. Minimum least squares regression across the six data points is used to show linearity, which is desired and expected.

#### 2.1.2. Acceleration

The ADXL345 (Analog Devices, Norwood, MA, USA) was chosen due to availability but is well suited to the application due to small size and low external part count. It has a 13-bit measurement resolution at ±16 g (3.9 mg/LSB), which allows measurement of inclination changes less than 1.0° [35]. The chip is supplied in a 3 mm × 5 mm × 1 mm, 14-lead land grid array (LGA) packaging and only requires three external capacitors [35]. Data are accessible through either Serial Peripheral Interface (SPI) or Inter-Integrated Circuit (I2C) digital interfaces [35]. Both interfaces are implemented in the custom PCB revisions integrated with strain gage conditioning circuitry, but the SPI version was ultimately chosen for this application because of its superior speed over I2C and the availability of source code for fast logging to the microSD card using SPI with the SDFat library [36]. The accelerometer is calibrated using the acceleration due to gravity [37,38]. To calibrate the sensor in a single direction, the sensor is rotated 180° so that it experiences a 2 g (−1 g to +1 g) step function [39]. The calibration was performed for each of the three axes with results as expected.

#### 2.1.3. Power Consumption

A constant power consumption of ~157 mW was measured using a Keithley (Solon, OH, USA) 2400 Source Measurement Unit (SMU). The power consumption is shown in Figure 7 where the current draw is ~47.7 mA at nominal 3.3 V. The documentation claims an average power draw of the ATSAMD21 microprocessor on board the Adafruit Feather M0 with regulator circuitry is 11 mA while both the red and green LED each draw 1 mA when lighted [40]. Since the circuit is modified to use 1000 Ω strain gages versus the 350 Ω strain gages of the reference design [29], the reduced current draw is calculated from 8.5 mA to 3 mA per channel. This would suggest that the strain conditioning components and accelerometer consume less than ~30 mA. The smallest available battery compatible with the Adaruit Feather M0 has a capacity of ~100 mAh, which would last for ~2 h given the ~47.7 mA current draw. This is sufficient time to conduct an in vivo or simulated experiment in a lab environment.

### 2.2. Device Enclosure and Attachment to the Rostrum

A “watch band” concept was developed to attach the device to the fish around the rostrum. The profile of the device is reduced as much as possible by using the battery sandwiched between the boards and the boards soldered together via header pins. Designed using a CAD program, a 3D print is made as a mold for the watch band and enclosure where the negative space for the device is also created by a 3D print. A 2-part mix of pourable silicone rubber is used to cast the band. Mold Star 15 SLOW Platinum Cure Silicone by Smooth On (Macungie, PA, USA) provided the best combination of flexibility and durability. The other silicone under consideration was Mold Star 31T Platinum Cure Silicone also by Smooth On. The main two differences between these two mixes are the Shore hardness and cure times. Mold Star 31T cures to a workable point in 25 min, but the end product is much less flexible than Mold Star 15. Therefore, Mold Star 15 was chosen to cast the band.

The two ends of the band are secured around the rostrum using 100% silicone all-purpose adhesive sealant (DAP, Baltimore, MD, USA). A cure time of 24 h is required to properly bond the two ends of the silicone band together, so the adhesion was performed on a 3D printed replica of a rostrum. Standard steel spring clamps were used to hold the bands together while curing. The resulting band is shown in Figure 8. The device is painted with Gagegkote 8 (Vishay, Malvern, PA, USA), wrapped with vinyl electrical tape #35 (3M, Saint Paul, MN, USA), then sprayed with Teflon^®^ silicone lubricant (Du Pont, Wilmington, DE, USA) before being inserted into the band enclosure. At this point, the wires connecting the strain gages to the device must be soldered in place. Finally, due to the fast cure time of Mold Star 31T, pouring the mixture into the band following insertion of the instrument provided a water tight seal that also bonded to the external wires.

### 2.3. Techniques for Strain Gage Attachment to Rostrum

Strain gages were attached to amputated rostrum using two techniques: (1) Dremel sanding of the skeleton, and (2) fasciotomy and Dremel sanding of the skeleton. Three different adhesives were used: (1) Permabond 240 alphacyanoacrylate ester (Permabond, Winchester, UK) (2) Vetbond (3M, Saint Paul, MN, USA), and (3) HBM Z70 cyanoacrylate (HBM, Darmstadt, Germany). Both techniques are comparatively minor procedures since the endoskeleton lies close to or is flush with the epithelial surfaces of the rostrum. Epithelium overlaying the skeleton is typically <3 mm thick and in many cases the skeleton is partly exposed. Because sanding takes several minutes; however, future in vivo work would necessitate life-support for the fish (i.e., mechanical flushing of the gills) and either anesthesia or topical application of analgesics to minimize pain. Handling procedures would be similar to those developed for endoscopic studies of sturgeon which are related to Paddlefish [41,42].

The successful attachment of wires to the gages indicated that both techniques and all three adhesives were strongly affixed, but Permabond 240 alphacyanoacrylate ester adhesive took longer to dry (Vetbond and HBM Z70 dried instantaneously). Preliminary results of 3 point bending of the specimen indicated that strongest responses were obtained from gages proximal to the applied force and gages not touching soft tissue (e.g., right rostral tip did not register due to dampening of signal by adjacent soft tissues). These tests suggested that sanding was not critical to getting a response, but that direct attachment to bone and complete exposure was. In addition, it was found that sodium bicarbonate Teflon^®^ tape (Du Pont, Wilmington, DE, USA) adhered to the strain gages. Kimwipes (Kimberly Clark, Irving, TX, USA) served best to hold the strain gage with glue. P120 type sand paper gave better results as compared to P240 in reaching the hard cartilage efficiently. This could be relevant to any subsequent work with live fish, since sanding of the skeleton will present challenges to Institutional Animal Care and Use Committee (IACUC) approval. All gages were attached directly to non-abraded cartilage of the rostrum: stellate lattice at tip, longitudinal rod at shaft. Attachment was done by fasciotomy (removal of skin and underlying tissue) using a #11 scalpel, lightly scraping (but not sanding) the surface of the skeleton, and affixing with Vetbond tissue adhesive No. 1469 (3M, Saint Paul, MN, USA)—an n-butyl cyanoacrylate. Soft tissues were thicker laterally and anteriorly on rostrum, and thinnest centrally and posteriorly on the rostrum. The type of glue used depends on the purpose of the measurement, conditions the specimen will endure, and the length of time the installation will need to last.

## 3. System Validation

The ultimate goal of this work is to collect in vivo measurement of the strain and acceleration experienced by the Paddlefish rostrum as the fish swims. However, the specific procedures for in vivo testing will require approval from the IACUC. Therefore, a proposed intermediate experimental approach is to collect strain and acceleration data experienced by amputated rostra under simulated swimming conditions by statically mounting them in a swim tunnel tank. This section describes the preliminary procedures of such an experiment where the purpose is to demonstrate and evaluate the efficacy of the instrumentation rather than purport any characteristics of the rostrum itself based on the collected measurements. The true reactions of the rostrum to water flow were not as important for this experiment as verification that the instrumentation process of attaching, securing, and recording is successful.

### 3.1. Methods and Materials

The test was performed in a 1200 L Brett-type swim tunnel, using a clear acrylic tubular insert (37 cm diameter by 151 cm long) situated between inflow and outflow collimators (aka flow filters) that create a column of rectilinear flow with minimal turbulence [43]. Figure 9 shows the entire swim tunnel apparatus with the swim test area and direction of flow noted.

The bracket to mount the rostrum within the tunnel was designed specifically for this experiment using 303 stainless steel as the primary material. Its basic structure consists of two shafts with four-inch center-to-center spacing and one-inch diameters. Both ends of each shaft provide 5/8” tapped holes allowing swivel-leveling feet to be inserted. These feet allow the mounting bracket to expand to fit firmly against the walls of the flume, and accommodate flumes with varying diameters. Using quick release shaft collars, a piece of angle iron is positioned between the two shafts. The amputated rostrum is attached to the extended end of the angle iron via a 3D printed clamp, and the quick release shaft collars allow the height and angle of the rostrum to be varied while in the flume.

The rostrum used in this experiment was taken from a fully-grown adult fish collected in the Mississippi River near Vicksburg, MS, USA. The rostrum weighed 375.3 g and was 367 mm in overall length. The standardized rostrum length, measured as a straight line from anterior rim of the eye’s orbit to the tip of the rostrum, was 353.41 mm, which corresponds to a Paddlefish of 650–950 cm eye-fork length and 4–16 kg body mass. The rostrum width was expanded anteriorly: 79.5 mm at the base, 78.8 mm at the shaft, but 104.0 mm at the tip. The rostrum depth was tapered anteriorly: 45.1 mm at the base, 18.6 mm at the shaft, and 13.8 mm at the tip. The strain gages were mounted dorsally and ventrally, 95.4 and 96.8 mm, respectively, posterior to rostrum tip, on the central cartilage. This rostrum had been previously used in a related experiment, so minimal surface preparation was required. Minor sanding was performed to remove old adhesive along with re-cauterizing the application area to prevent surface moisture from pooling. The HBM Z70 was chosen as the adhesive due to its fast curing and high viscosity, which provides a more uniform layer under the strain gage. In order to guarantee a fast cure and strong adhesion, HBM BCY01 accelerant was applied to the back of the strain gage substrate and allowed to fully dry. M-Coat FB-2 Butyl Rubber Sealant (Vishay, Malvern, PA, USA) placed over the strain gage provided a barrier to keep water from penetrating. A continuous bead of HBM Z70 around the perimeter seals the M-Coat FB-2 in case surface abnormalities prevent good adhesion. The Vishay model 134-AWP hookup wire used to connect the strain gages to the datalogger run the length of the rostrum. HBM Z70 and HBM BCY01 placed along the entire length of the cabling adds strain relief to prevent the turbulence of the water from damaging the cables or strain gage. The fully instrumented rostrum is shown in Figure 10.

Finally, the instrumented rostrum, in its mount, was positioned at the rear of the tube so that the tip of the rostrum was in the center of the tube pointing into the flow (Figure 11). This set-up simulates a fish swimming at uniform depth upriver, but could be adjusted upward (approximately 30°) to simulate a fish rapidly surfacing, or downward (approximately 20°) to simulate a fish gradually diving. Flow was controlled using a digital hand unit set for a slow (approximately 15 cm/s) or fast speed (approximately 75 cm/s). Data were collected continuously for 5-min from a horizontally-positioned rostrum at three discrete intervals corresponding to static, slow- and fast-flowing water. After testing was completed, the rostrum was removed from the tank. The strain gage wires were cut and the entire watchband was removed from the rostrum. The recording device was then removed from the watchband enclosure by cutting through the silicone.

### 3.2. Results

The instrumentation successfully collected data continuously from the horizontally-positioned rostrum for three discrete 5-min intervals corresponding to static, slow-, and fast-flowing water. The microSD card was removed from the device and inserted into a computer to post-process and analyze the data. The data were recorded on the microSD card as a comma separated value (csv) file for each test. At this time, it is not possible to reuse the device due to excess silicone preventing the reinsertion of the microSD card. The algorithm to convert the recorded signal to strain was presented in [28] and was followed here. The average of the recorded voltages of the static flow test was used as the initial conditions for all of the respective strain calculations to allow for a direct comparison.

The first analysis method consists of calculating statistics (means, standard deviations). This was performed over 96,000 measurements encompassing each of the five minute tests respectively for strain (Table 1) and acceleration (Table 2). Secondly, the time series of the strain data were observed for temporal variation in sensor reading (Figure 12). Towards the consideration of sensor drift and in light of observed anomalies in the time series data, statistics (means, standard deviations) were also calculated over 64,000 strain measurements representing the final 200 s of each of the tests, respectively, removing the first 100 s of data (Table 3). For the same reason, time-based minimum least squares regressions across a subset of 80,000 data points representing the final 250 s of each test were calculated, removing the first 50 s of data (Table 4).

The statistical data suggest that ventral strains are appreciably greater than dorsal strains, particularly in slow flow, but that strain at high flow is only marginally greater than at slow flow (Table 1 and Table 3). Accelerations in all three directions were comparatively low and did not change appreciably with flow rate, but variation in measurements were greater at the highest flow (Table 2) suggesting a higher amplitude vibration at the higher flow. Differences between the dorsal responses and ventral responses to slow water after 100 s (2x increase in mean strain vs. 10X increase in mean strain, respectively) may reflect a fundamental difference in dorsal and ventral endoskeletal structure (Table 3). The dorsal lattice includes buttress-like supports that connect the cranium of the fish to the rostrum’s axial cartilage approximately 25% distal to the base of the rostrum [44]. These buttress-like rods of cartilage may function like flying buttresses in human-constructed architecture [45], the lateral elements transferring stress forces (i.e., hydraulic strains on the rostrum) to an object of greater mass (i.e., the body of the Paddlefish). Lastly, the absence of a concordant response in strain in higher water velocity (i.e., little or no change in strain despite a 5x increase in flow from slow to fast) underscores the effectiveness of this unique structure in effective energy-dispersion [44].

The time series of the strain data (i.e., bi-variate plots of dorsal and ventral strains over duration of test) were observed for temporal variation in sensor readings (Figure 12). An initial sharp increase followed by the approach to near horizontal asymptote within ~50 s was generally observed in all cases. This behavior is suggested to be resulting from transient electronic effects during startup. Since this anomaly consisted of >15% of test duration, it was determined that it would be necessary to conduct longer duration testing (approaching the targeted ~2 h in vivo duration) to draw any conclusions about the presence of sensor drift and any impact this would have on drifts in the bending moment and axial force calculations on the rostrum cantilever. This is left as future work.

The asymptotic region is further considered by removing the first 50 s and calculating the strains of each test using the average of the recorded voltages from each test as respective initial conditions, effectively removing any offset from zero. This results in near-zero mean (or DC filtered) signals for each test on which respective time-based minimum least squares regressions across the remaining 80,000 data points have been calculated (Table 4). Declines at higher water velocity in r-square of the time-based regressions (e.g., 0.68 to 0.02 dorsally, and 0.80 to 0.45 ventrally) reflect the increased spread of points above and below the regression lines with increased water velocity and demonstrate that the gages can detect turbulence-generated strain associated with higher flows (Table 4). Positive slopes for 5/6 graphs suggest that slight drift takes place that is not associated with the 50 s start-up anomaly. Decreased slope of the time-based regressions in flowing water suggests rapid and effective dispersion of energy over time.

## 4. Conclusions

This work describes and tests an instrumentation system designed to measure the strain and acceleration experienced by the Paddlefish rostrum as the fish swims. In order to not encumber the swimming motion with wires, a low-SWaP electronic data logger is implemented with a silicone watch band style enclosure fitting around the rostrum. The measurement capability has been validated independently in lab experiments and within an integrated system by simulated swimming conditions on an amputated rostrum. Ultimately, this device may be used for in vivo testing of live Paddlefish (or other species of fish with prominent rostra). Such tests would be challenging using free-swimming Paddlefish in the field, however, due to the extensive high-speed movements and physically complex habitats of any single fish. Recent swimming and metabolic studies using field-collected Paddlefish confined to a mobile-swim tunnel, however, make in vivo tests a realistic possibility.

## Figures and Tables

**Figure 1 biosensors-10-00037-f001:**
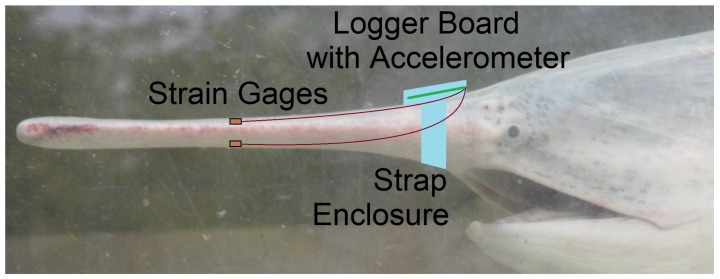
Proposed placement of instruments on the rostrum of a Paddlefish for future in vivo experiments. This is the same configuration used for the proof-of-concept experiment with an amputated rostrum described in this paper.

**Figure 2 biosensors-10-00037-f002:**
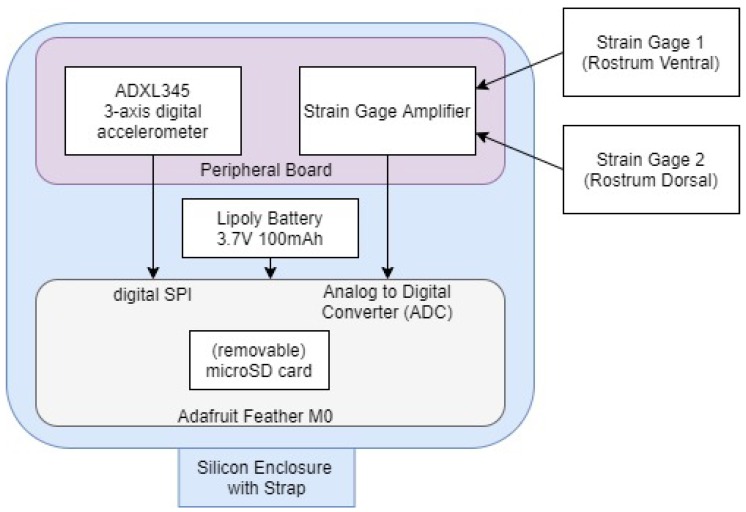
Instrumentation design concept.

**Figure 3 biosensors-10-00037-f003:**
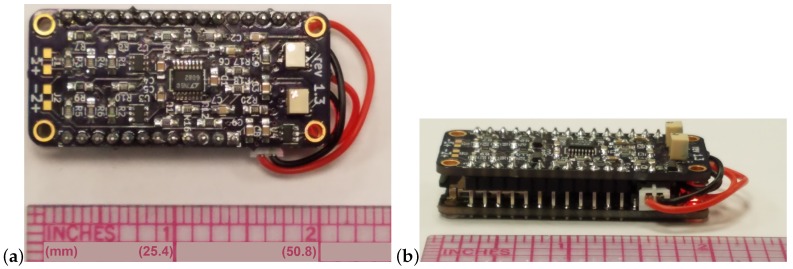
Data logger: bird’s-eye view (**a**) and eye-level view (**b**).

**Figure 4 biosensors-10-00037-f004:**
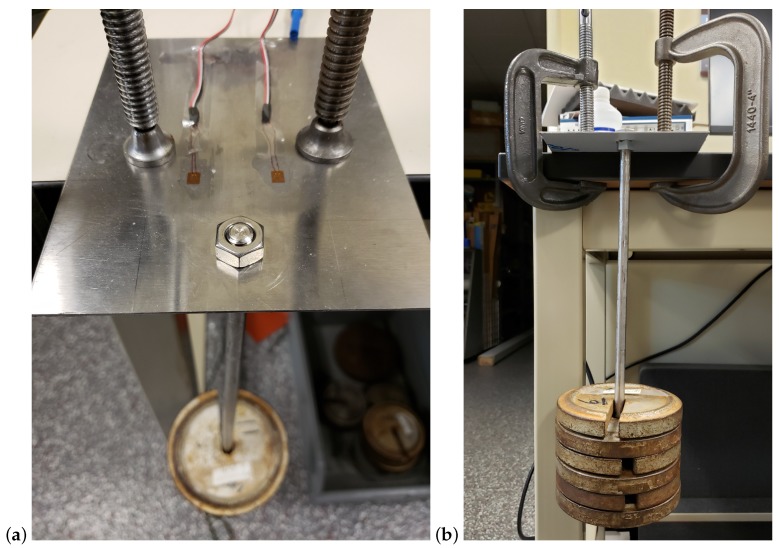
Load cell calibration test setup: oblique bird’s-eye view (**a**) and eye-level view (**b**).

**Figure 5 biosensors-10-00037-f005:**
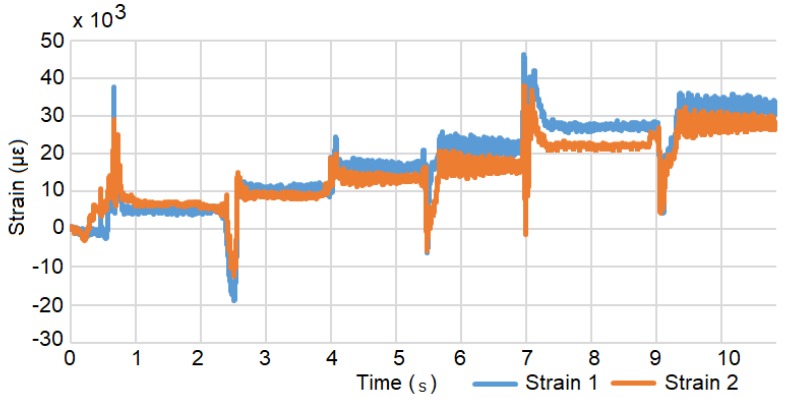
Strain versus time during the load test. The step response occurs when a weight is added.

**Figure 6 biosensors-10-00037-f006:**
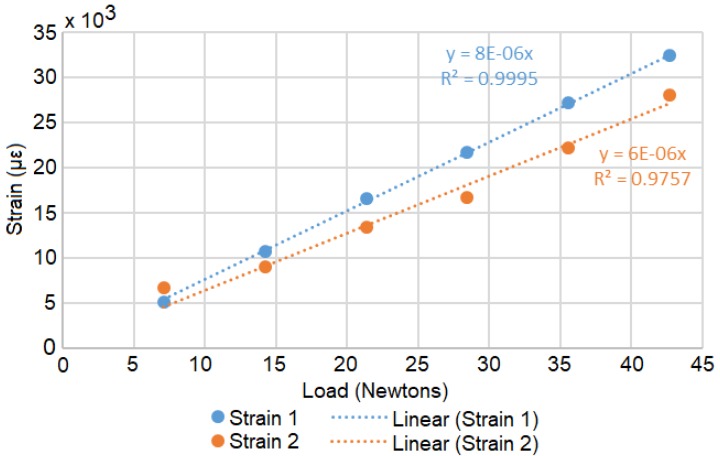
The processed average loads with minimum least squares regression equations.

**Figure 7 biosensors-10-00037-f007:**
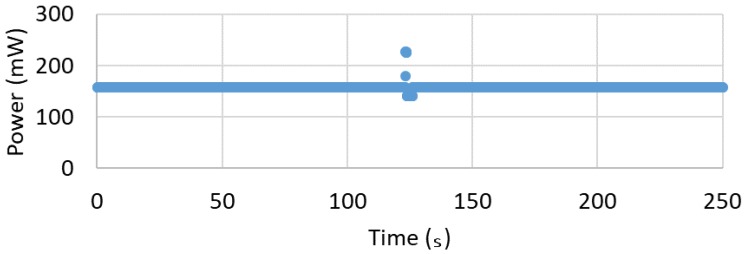
Power versus Time shows consistent 157 mW excepting an anomalous spike which is currently unaddressed.

**Figure 8 biosensors-10-00037-f008:**
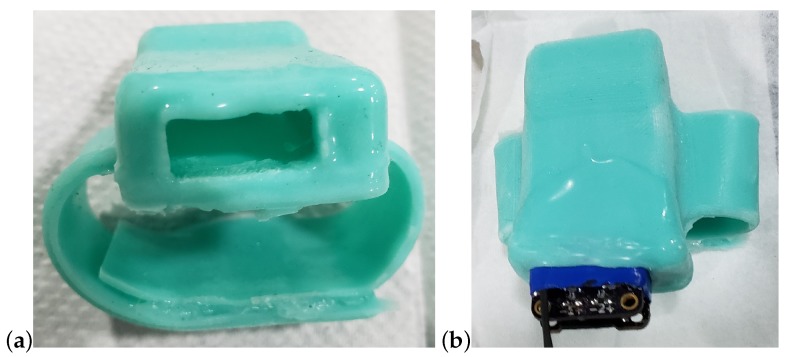
Watch-band type enclosure shown (**a**) with opening to insert the device, and (**b**) the partially inserted device, ready to solder the wires.

**Figure 9 biosensors-10-00037-f009:**
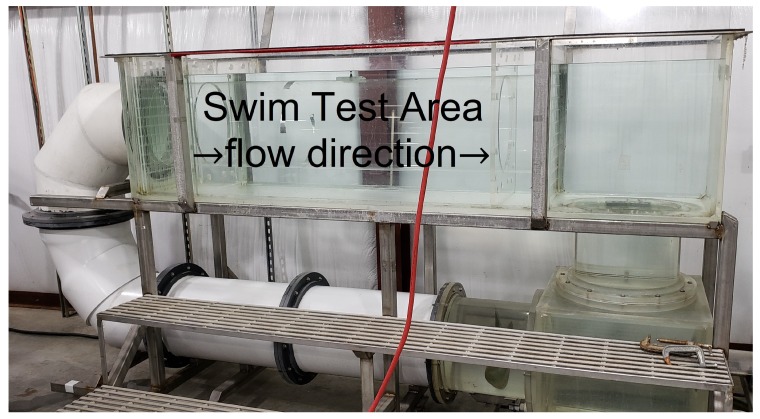
Brett-type swim tunnel used in proof-of-concept experiment. The orange power cord is visible in the foreground.

**Figure 10 biosensors-10-00037-f010:**
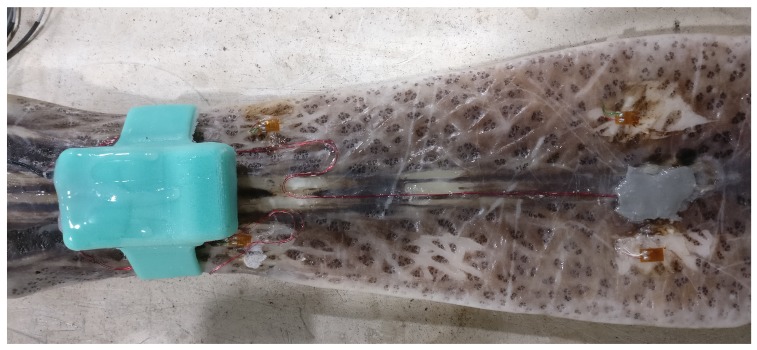
Bird’s-eye view of rostrum with instrumentation. Rostrum oriented posterior-anterior left-right. Dorsal strain gage is under the gray sealant, accelerometer, and data logger inside the turquoise silicone “watch band.”

**Figure 11 biosensors-10-00037-f011:**
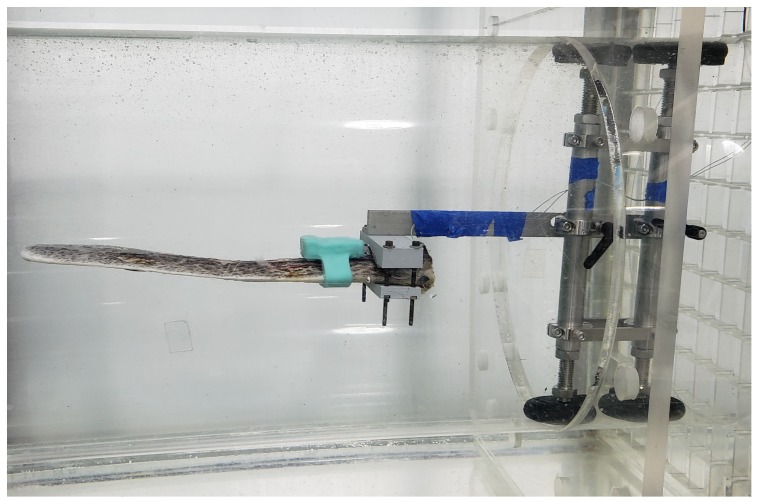
Eye-level view of rostrum mounted in the swim tunnel during testing.

**Figure 12 biosensors-10-00037-f012:**
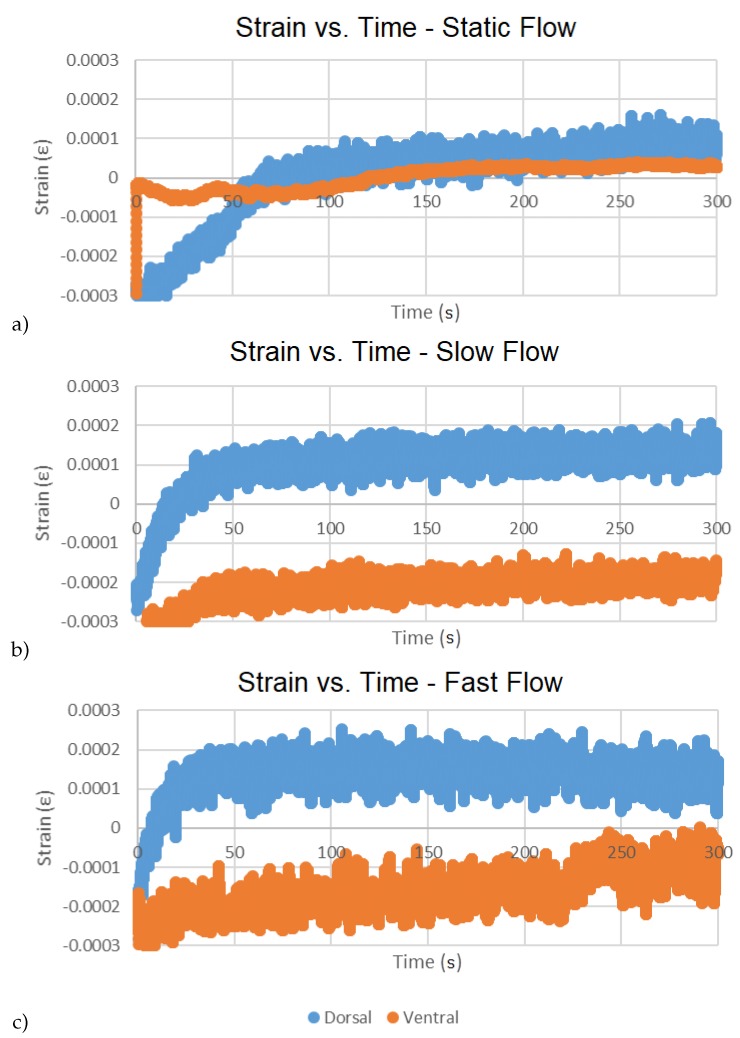
Time series bi-variate plots of dorsal and ventral strains over duration of tests for respective flows: static (**a**), slow (**b**), fast (**c**).

**Table 1 biosensors-10-00037-t001:** Statistics (means, standard deviations) of strain measured on dorsal and ventral gages of the rostrum during horizontal placement in three flows are calculated over 96,000 measurements encompassing the full duration of each of the five minute tests, respectively.

	Dorsal Strain (ϵ)		Ventral Strain (ϵ)	
flow	mean (μ)	std.dev. (σ)	mean (μ)	std.dev. (σ)
static	1.46 × 10${}^{-8}$	1.2024 × 10${}^{-4}$	2.24 × 10${}^{-9}$	4.703 × 10${}^{-5}$
slow	9.75 × 10${}^{-5}$	8.8432 × 10${}^{-5}$	-2.2× 10${}^{-4}$	4.9406 × 10${}^{-5}$
fast	1.34 × 10${}^{-4}$	7.9761 × 10${}^{-5}$	-1.7× 10${}^{-4}$	6.4527 × 10${}^{-5}$

**Table 2 biosensors-10-00037-t002:** Statistics (means, standard deviations) of three-axis accelerations of the rostrum during horizontal rostrum placement in three flows are calculated over 96,000 measurements encompassing the full duration of each of the five minute tests, respectively.

	*x*-Direction (*g*)		*y*-Direction (*g*)		*z*-Direction (*g*)	
flow	mean (μ)	std.dev. (σ)	mean (μ)	std.dev. (σ)	mean (μ)	std.dev. (σ)
static	2.852 × 10${}^{-2}$	4.10 × 10${}^{-3}$	−4.772 × 10${}^{-2}$	3.09 × 10${}^{-3}$	−1.00442	3.36 × 10${}^{-3}$
slow	2.291 × 10${}^{-2}$	3.91 × 10${}^{-3}$	−4.702 × 10${}^{-2}$	4.35 × 10${}^{-3}$	−1.00441	3.46 × 10${}^{-3}$
fast	2.65 × 10${}^{-2}$	1.132 × 10${}^{-2}$	−4.574 × 10${}^{-2}$	1.319 × 10${}^{-2}$	−1.00554	1.488 × 10${}^{-2}$

**Table 3 biosensors-10-00037-t003:** Statistics (means, standard deviations) of strain measured on dorsal and ventral gages of the rostrum during horizontal placement in three flows are calculated over 64,000 measurements encompassing the final 200 s of each test, respectively (removing the first 100 s of the full duration).

	Dorsal Strain (ϵ)		Ventral Strain (ϵ)	
flow	mean (μ)	std.dev. (σ)	mean (μ)	std.dev. (σ)
static	5.83463 × 10${}^{-5}$	2.82712 × 10${}^{-5}$	2.02928 × 10${}^{-5}$	1.50277 × 10${}^{-5}$
slow	1.24807 × 10${}^{-4}$	2.21648 × 10${}^{-5}$	−2.00395 × 10${}^{-4}$	2.11473 × 10${}^{-5}$
fast	1.47701 × 10${}^{-4}$	3.01531 × 10${}^{-5}$	−1.43745 × 10${}^{-4}$	4.94369 × 10${}^{-5}$

**Table 4 biosensors-10-00037-t004:** Time-based minimum least squares regressions of final 250 s of strain measured on dorsal and ventral gages of the rostrum during horizontal placement in three flows (removing the first 50 s of the full duration).

	Dorsal Strain			Ventral Strain		
flow	slope	intercept	r-square	slope	intercept	r-square
static	5 × 10${}^{-7}$	1 × 10${}^{-4}$	0.6806	3 × 10${}^{-7}$	7 × 10${}^{-5}$	0.8006
slow	1 × 10${}^{-7}$	3 × 10${}^{-5}$	0.1652	2 × 10${}^{-7}$	3 × 10${}^{-5}$	0.2375
fast	−6 × 10${}^{-8}$	1 × 10${}^{-5}$	0.0202	5 × 10${}^{-7}$	9 × 10${}^{-5}$	0.4516

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
