# Peer review of "Instrumenting Polyodon spathula (Paddlefish) Rostra in Flowing Water with Strain Gages and Accelerometers"

_biosensors, 2020, doi:10.3390/bios10040037_

Round 1
Reviewer 1 Report
This is an interesting and well-written manuscript about the development of a biosensor capable of measuring strain and acceleration at the rostrum of paddlefish. The manuscript reports the methodology to fabricate the biosensor, which includes the development of the amplifier circuit, a description of the accelerometer and strain gauges, software and device enclosure, and attachment to the rostrum. This is an interesting piece of work suitable for Biosensors. The results are clear and properly discussed. The following comments should be addressed before the manuscript can be recommended for publication:
Abstract
-The abstract should be shortened; it is too long (around 300 words). (It should be between 150 and 200 words).
Introduction
-The authors should discuss other in vivo measurements in other animals where acceleration and/or strain is measured.
-The authors should discuss other non-in vivo measurement in animals or part of animals since the results presented in this paper are not in vivo.
Section 2
Please indicate in Figure 1 the position of the accelerometer.
In general, most of the graphs with data need improvement (Fig. 9, 10, 12, 13, 15-18, A6). They should look more professional. The style of all graphs should be homogenized, i.e., similar colors, font, font size, size, labels, frame, etc.
Reviewer 2 Report
Thank you for submitting this novel and interesting manuscript for review. Such measurements are challenging, and you have clearly given much attention to details of your electrical design.
This paper is in my opinion however more appropriate for publication as a technical note, at this stage of your work, because of the many technical details included and lack of study results in vivo; you could succinctly precis some of these details without loss of important content to improve readability.
In this application, where you are wanting to carry out strain measurements underwater, I think you need to include more appropriate tests to ascertain the reliability of your electronics and gauges in a wet environment. Such measurements are challenging and need extensive testing before being relied upon.
Although you have carried out calibration of the gauges on metal, the measured strains in vivo will depend on the orientation of the gauges placed. Also I don't think you have tested the entire system together yet, unless I missed that. It would also be nice to see a fully encapsulated measurement system with gauges attached to plastic or bone, for underwater testing and long term strain measurements.
I could find no reference as to how you intend to telemeter the data from the fish?
L31-32. Fish (pl) is fish, not fishes.
The last paragraph of the Introduction (52-55) belongs at the start of the following section.
Your amplifier circuit uses two cascaded op amps (a x1 differential amp followed by a x501 single-ended). Why not split the gain more evenly (22.4) which would give you better GBW efficiency and noise immunity. Also it is easier to offset the bridge imbalance without having to use precision resistors.
L128-147 and L159-173: these sections have a lot of detail which is hard to follow and not appropriate for this paper; it would fit better in a technical report/paper, and could be simply summarised here. The ADC is not prior mentioned; where/what is this? An overall circuit topology would be helpful.
2.3 Accelerometer: what is the purpose of this? Needs to be discussed.
L186-206 There is no mention of underwater testing to find the most suitable adhesive, or of the use of ISO10993-5/6 adhesives for biocompatibility. Your tests were done on cadaveric bone (wet or dry?); wet bone will be very difficult to bond gauges to without drift for long periods.
3.2 This is basic data from the accelerometer and can be omitted.
Future work: I assume you intend to carry out in vivo trials inside a fish tank; will this adequately investigate the science in an appropriate setting?
With a current draw of 50mA, you will not be able to power the device for more than a few days without a heavy battery, but if the tests are to be carried out in the lab perhaps this does not matter.
This article gives no details of work carried out to package the electronics and gauges appropriately for underwater use, and so this will need to be done before moving to an in vivo study. Will you use radio transmission?
Best wishes for your future work and eventual experimental study.
Reviewer 3 Report
The main issues are the following:
Authors said: “we suggest that the models proposed by other authors can be further informed by data collected during in vivo experiments…”. However, they didn´t do in vivo testing.
Authors said: “…the novelty of their work is framed as a low-size, weight, and power and waterproof package instrumentation challenge”. Is this a novelty? In my opinion, the instrumentation used is well-known. Apart from this, no more contribution is reported.
The conclusion section is missing.
Round 2
Reviewer 2 Report
Thank you for sending this revised manuscript so promptly. Most areas of my original concerns have been addressed, and I think your revision is much improved. However I would prefer that you include in your results the strain data over the length of time for which you propose to carry out the in vivo measurements. Did the strains drift and by how much? What does this translate into for drift in the bending moment and axial force on the rostrum cantilever, and how significant is this in the context of the study?
The drift data, especially if measured over several sessions and for extended periods, will validate your encapsulation techniques and give your in vivo data more credibility. I still think that you don't need to include data regarding calibration of the accelerometer, if using a MEMS device, but I leave that to you to decide.
Reviewer 3 Report
1) Please, remove references in the conclusion section.
2) If possible, remove the 3D CAD models. I think they are less important.
3) The title said, "..in Motion.." Is it true?
4) In the conclusion section, remove " ...as the fish swims".
